# Reducing greenhouse gas emissions from pig slurry by acidification with organic and inorganic acids

**Frederik R. Dalby◉\*, Lise B. Guldberg, Anders Feilberg, Michael V. W. Kofoed**

Department of Biotechnology and Chemical Engineering, Faculty of Technical Science, Aarhus University, Aarhus, Denmark

\* fd@bce.au.dk

**Data Availability Statement:** All relevant data are within the manuscript and its Supporting Information files.

**Funding:** This work was funded by the Danish Agricultural Agency under the Ministry of

## Abstract

Methane ($CH_4$) emission from pig slurry is a large contributor to the climate footprint of livestock production. Acidification of excreta from livestock animals with sulfuric acid, reduce $CH_4$ emission and is practiced at many Danish farms. Possible interaction effects with other acidic agents or management practices (e.g. frequent slurry removal and residual slurry acidification) have not been fully investigated. Here we assessed the effect of pig slurry acidification with a range of organic and inorganic acids with respect to their $CH_4$ inhibitor potential in several batch experiments (BS). After careful selection of promising $CH_4$ inhibitors, three continuous headspace experiments (CHS) were carried out to simulate management of manure in pig houses. In BS experiments, more than <99% $CH_4$ reduction was observed with $HNO_3$ treatment to pH 5.5. Treatments with $HNO_3$, $H_2SO_4$, and $H_3PO_4$ reduced $CH_4$ production more than acetic acid and other organic acids when acidified to the same initial pH of 5.5. Synergistic effects were not observed when mixing inorganic and organic acids as otherwise proposed in the literature, which was attributed to the high amount of acetic acid in the slurry to start with. In the CHS experiments, $HNO_3$ treatment reduced $CH_4$ more than $H_2SO_4$, but increased nitrous oxide ($N_2O$) emission, particularly when the acidification target pH was above 6, suggesting considerable denitrification activity. Due to increased $N_2O$ emission from $HNO_3$ treatments, $HNO_3$ reduced total $CO_2$-eq by 67%, whereas $H_2SO_4$ reduced $CO_2$-eq by 91.5% compared to untreated slurry. In experiments with daily slurry addition, weekly slurry removal, and residual acidification, $HNO_3$ and $H_2SO_4$ treatments reduced $CO_2$-eq by 27% and 48%, respectively (not significant). More cycles of residual acidification are recommended in future research. The study provides solid evidence that $HNO_3$ treatment is not suitable for reducing $CO_2$-eq and $H_2SO_4$ should be the preferred acidic agent for slurry acidification.

## Introduction

Methane ($CH_4$) emission from management of livestock manure contributes approximately 6% of global anthropogenic $CH_4$ emission [1] and is a considerable source of global climate

Environment and Food, Denmark (grant no. 33010-NIFA-19-725). The funder provided support in the form of salaries for authors Frederik R. Dalby, Lise B. Guldberg, Anders Feilberg, and Michael V.W. Kofoed but did not have any additional role in the study design, data collection and analysis, decision to publish, or preparation of the manuscript. The specific roles of these authors are articulated in the 'author contributions' section."

**Competing interests:** The authors have declared that no competing interests exist.

change [2]. Methane emission factors are higher for pig slurry than cattle slurry [3] and the potential for mitigation is greater in pig slurry management systems. In many finisher pig production facilities, the slurry is stored underneath the slatted floor in pits or channels and kept there throughout the production cycle [4]. In northern regions, temperatures are lower in outside storage tanks and the rate of $CH_4$ production per ton slurry will therefore decrease significantly if slurry from pig houses is pumped to external storages. Frequent slurry removal from the pig houses by gravimetric emptying, scraping, or flushing, therefore holds a $CH_4$ reduction potential, but such practices are carried out in <1% of pig production facilities [4]. Another promising mitigation technology is slurry treatment with acidic agents. This approach was implemented to reduce ammonia emission as it shifts the acid-base equilibrium towards ammonium ions ($NH_4^+$) that are nonvolatile [5]. Lowering the slurry pH also reduces $CH_4$ emission, but the mode of inhibition is not completely understood. Common practice is to lower the pH to 5.5 with $H_2SO_4$ with reported $CH_4$ reductions of 63–99% [6]. Proposed inhibition mechanisms include $H_2S$-mediated inhibition [7], which is mainly derived from sulfate reduction [8], uncoupling of the cell membrane by protonated fermentative products [9, 10], or competitive inhibition of methanogens by microorganisms able to respire more energetic electron acceptors [11]. The latter may be relevant if acidification is carried out using $H_2SO_4$ or $HNO_3$, which dissociates to the more preferable electron acceptors, sulfate and nitrate, than $CO_2$ during methanogenesis [11]. Nitrate is a highly preferable electron acceptor in the absence of oxygen and it may be partially reduced by microorganisms to $N_2O$ or fully reduced to $N_2$ in a cascade of reactions termed denitrification [12, 13]. Nitrous oxide is a greenhouse gas 273 times more potent than $CO_2$ over 100 years [14]. It may also be produced as a side product of $NH_3$-oxidation under semi-aerobic conditions when $NH_4^+$ is oxidized to $NO_2^-$, but this process is considered nearly absent in slurry channels and storages without crusts [12].

As $NO_3^-$ is a preferable electron acceptor, $CH_4$ potential may be significantly reduced due to competition for substrates by denitrifying bacteria if treated with $HNO_3$. Large $CH_4$ reductions were previously observed from $HNO_3$ acidified cattle slurry, but simultaneous increases in $N_2O$ emission attributed to denitrification activity were also detected [15, 16]. It appears difficult to avoid $N_2O$ emission from sole $HNO_3$ treatment, but combining acids might generate synergistic effects. Lately, $CH_4$ emission from slurry was reduced by combining $H_2SO_4$ with acetic acid as compared to pure $H_2SO_4$ treatment to pH 5.5 [6]. This effect was possibly related to protonated acetic acid uncoupling the cell membrane [9, 10] and draws attention to the idea of combining organic and inorganic acids to achieve synergistic mitigation effects. Other organic acids have been examined as replacements for $H_2SO_4$, but with a focus on $NH_3$ emission and manure characteristics [17, 18].

Acidification strategies must be optimized and integrated with other management practices such as frequent slurry removal from slurry channels inside animal houses to external storages. Frequent slurry removal leaves a residual slurry fraction that may function as an inoculum that speeds up the anaerobic processes in newly produced slurry resulting in accelerated $CH_4$ production [19]. However, complete removal of residual slurry is impractical at large farms [20] or in farms with typical designs of pits and slurry channels. Coupling acidification of the residual slurry with frequent slurry removal may enhance the total mitigation potential and reduce acidification expenses as compared to bulk acidification. Laboratory tests of residual slurry acidification have shown very promising results a with reduction of $CH_4$ emission of up to 99% over 116 days [21]. In pilot-scale storage tanks (10 $m^3$), 2.12 $m^3$ residual slurry which was acidified before addition of 8.48 $m^3$ fresh slurry reduced $CH_4$ emission by 77% over 155 days compared to a control case where the residual slurry was not acidified before the addition of fresh slurry [20]. Neither of the studies examined residual slurry acidification in a setup with continuous or daily addition of slurry, which could potentially fuel $CH_4$ emissions. Further,

the studies focused on cattle slurry in outside storages, which has a lesser potential of $CH_4$ emission than pig slurry stored inside a pig house [3].

In this study, the aim was to determine the optimal acid treatment for reducing greenhouse gas emissions from pig slurry and to understand how acid treatment efficiency depends on dosage and frequent removal of slurry. This was accomplished in two steps; 1) Screening of $CH_4$ production from pig slurry incubated with organic acids, inorganic acids, and combinations thereof in a simple assay using closed batch vials. 2) Following initial screening, we selected the most promising greenhouse gas mitigating acids for further investigations in a continuous gas monitoring setup, designed to simulate frequent slurry removal and residual slurry acidification in a pig pen. This novel experimental approach gives a realistic indication of the mitigation technology potential in pig houses. Gas emissions were measured using state-of-the-art cavity ring-down spectrometry to capture temporal evolution of gas emission and its response to treatment and slurry management strategy. The work conducted here clearly indicated that $H_2SO_4$ should be the preferred acidic agent for slurry acidification, independently of the management practice.

## Materials and methods

Two types of incubation experiments were conducted in this study. These are referred to as batch screening experiments (BS experiment A, B, and C) and continuous headspace experiments (CHS experiment A, B, and C). A simplified overview of the experimental setups is shown in Fig 1. The batch screening experiments were done first to choose the most promising acidic agents for the more complex continuous headspace experiments. In the batch screening experiments the target pH was 5.5 since this is the target pH of commercial in-house acidification systems [5]. The methodology and procedures are detailed in the following sections.

### Slurry characterization

Slurry dry matter was measured as the slurry mass remaining after heating at 105˚C for 24 h in a B180 Oven (Nabertherm). The pH was measured with a Portamess module (Knick) and a DJ114 pH electrode. For $NH_4^+$ concentration, the slurry was pre-diluted 10–50 times with ultrapure water and measured with an ammonium test kit 1.00683 (Merck) on a Spectroquant NOVA 60 (Merck).

### Site description and slurry collection

**Experimental pig facility.** Pig slurries for all incubation experiments were obtained from an experimental pig facility at Aarhus University (Foulum, Denmark). There were two pens in each section and 15 finishing pigs (30–110 kg) per pen. Batch time for the pigs was 77 days and

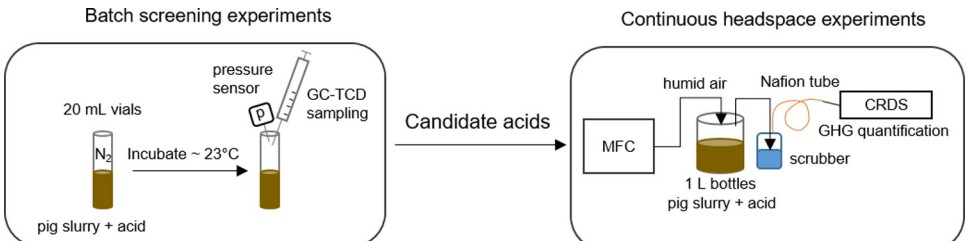

**Fig 1. Experimental overview of batch screening experiments and continuous headspace experiments.** GC-TCD is gas chromatography with a thermal conductivity detector, MFC is mass flow controller, CRDS is cavity ring-down spectrometry, GHG is greenhouse gas.

**Table 1. Description and characteristics of slurries.**

| Description | Applied in experiments | pH | Dry matter (%) | TAN[a] (g/L) |
|---|---|---|---|---|
| Vacuum flushed slurry from control section, sampled 2-7-2020 | BS[b] experiments and CHS[c] experiment A | 6.96 | 4.85 | 2.65 |
| residual slurry from control section, sampled 6-11-2020 | CHS experiments B and C | 7.21 | 9.35 | 3.11 |
| vacuum flushed slurry from experimental section, sampled 26-11-2020 | CHS experiment C | 6.66 | 8.93 | 2.07 |
| vacuum flushed slurry from experimental section, sampled 03-12-2020 | CHS experiment C | 6.77 | 7.76 | 2.17 |

[a] Total ammonia nitrogen.

[b] Batch screening experiments.

[c] Continuous headspace experiments.

batch days were scheduled in parallel in the two sections. In each pen, there was a 60 cm deep slurry pit below the slatted floor. The slurry pits were emptied with vacuum flushing leaving approximately 4 cm of residual slurry in the pit after vacuum flushing. The slurry management differed in the two sections: In the experimental section, the slurry was flushed every 7th day whereas, in the control section, the slurry was flushed on day 40 and at the end of the batch on day 77.

**Slurry sampling.** An overview of applied slurry types is presented in Table 1. Vacuum flushed slurry for the batch experiments and CHS experiment A was sampled from the vacuum flushing system in the control section at day ~ 40 in a batch of pigs. Due to decreasing methanogenic activity in the vacuum flushed slurry a residual slurry with higher methanogenic activity was used in CHS experiments B and C to ensure that potential differences in treatments were still detectable. Residual slurry was sampled from the bottom of the pits in the control section on the day after the pigs were taken out of the section and after the pits were emptied by vacuum flushing. The residual slurry was sampled by scraping the 3–5 cm residual slurry layer on the pit bottom with the beaker and the slurry was stored in PFTE containers at 5˚C until further experimental treatment. For diurnal slurry addition in CHS experiment C, weekly vacuum flushed slurry was sampled from the experimental section and stored at 5˚C between daily additions to the incubation flasks. New vacuum flushed slurry was collected from the experimental section weekly to ensure the slurry was no more than 7th days old when applied in the experiments.

## Batch screening experiments

Vacuum flushed slurry was agitated with a magnet stirrer (Heidolph, MR 3001 K) in a 5 L bucket while pH was measured with a pH sensor (Knick) and samples were collected for later slurry characterization. Eight grams of the slurry was transferred to 20 mL serum vials while still agitating the slurry using a 10 mL pipette with cut tips to avoid clogging. The headspaces of the serum flasks were flushed with nitrogen then crimped with aluminum caps and incubated for 24 h at room temperature (~23˚C). After 24 h the vials were opened and treated with inorganic or organic acids alone (BS experiment A), inorganic acids combined with acetic acid (BS experiment B), and organic acids (BS experiment C). Organic acids included acetic acid, lactic acid, propanoic acid, citric acid, and formic acid whereas inorganic acids used were hydrochloric acid, sulfuric acid, nitric acid, and phosphoric acid. Details of treatment dose and BS experiment design are presented in Table 2. After acid treatment, the vials were flushed with nitrogen, crimped with aluminum caps, and incubated at room temperature. On a weekly or biweekly basis, the pressure and $CH_4$ gas concentration in the serum vials headspaces were measured using a pressure sensor fused with a needle and a gas chromatograph with a thermal conductivity detector (GC-TCD). After pressure measurement or gas sampling for GC-TCD

**Table 2. Treatment details of batch screening experiments.**

| Treatment | n | Dose (pH or millimolar (mM)) |
|---|---|---|
| BS experiment A, length: 6 or 18 days, Fig 5A | | |
| Untreated | 2 | – |
| Sulfuric acid | 3 | To pH 5.5 |
| Phosphoric acid | 3 | To pH 5.5 |
| Acetic acid | 3 | To pH 5.5 |
| Hydrochloric acid | 3 | To pH 5.5 |
| Nitric acid | 3 | To pH 5.5 |
| Lactic acid | 3 | To pH 5.5 |
| BS experiment B, length: 9 days, Fig 5B | | |
| Untreated | 3 | – |
| Acetic acid + nitric acid | 3 | 42 mM + to pH 5.5 |
| Acetic acid + sulfuric acid | 3 | 42 mM + to pH 5.5 |
| Acetic acid + phosphoric acid | 3 | 42 mM + to pH 5.5 |
| Acetic acid | 3 | 42 mM |
| Nitric acid | 3 | To pH 5.5 |
| Sulfuric acid | 3 | To pH 5.5 |
| Phosphoric acid | 3 | To pH 5.5 |
| BS experiment C, length: 7 days, Fig 5C | | |
| Untreated | 2 | – |
| Acetic acid + sulfuric acid | 1 | 21 mM + to pH 5.5 |
| Acetic acid + sulfuric acid | 1 | 60 mM + to pH 5.5 |
| Acetic acid + sulfuric acid | 1 | 89 mM + to pH 5.5 |
| Propionic acid + sulfuric acid | 1 | 23 mM + to pH 5.5 |
| Propionic acid + sulfuric acid | 1 | 53 mM + to pH 5.5 |
| Propionic acid + sulfuric acid | 1 | 82 mM + to pH 5.5 |
| Citric acid + sulfuric acid | 1 | 13 mM + to pH 5.5 |
| Citric acid + sulfuric acid | 1 | 24 mM + to pH 5.5 |
| Citric acid + sulfuric acid | 1 | 36 mM + to pH 5.5 |
| Lactic acid + sulfuric acid | 1 | 28 mM + to pH 5.5 |
| Lactic acid + sulfuric acid | 1 | 70 mM + to pH 5.5 |
| Lactic acid + sulfuric acid | 1 | 112 mM + to pH 5.5 |
| Formic acid + sulfuric acid | 1 | 30 mM + to pH 5.5 |
| Formic acid + sulfuric acid | 1 | 46 mM + to pH 5.5 |
| Formic acid + sulfuric acid | 1 | 60 mM + to pH 5.5 |
| Sulfuric acid | 3 | To pH 5.5 |

analysis, the vials were flushed with nitrogen and crimped with aluminum caps. The BS experiments were terminated after 1–3 weeks depending on the gas production rate.

## Continuous headspace experiments

Vacuum flushed or residual slurry was thoroughly stirred in a 5 L bucket with a large spatula while measuring the pH with a Portamess pH sensor (Knick) and sampling for later characterization of the slurry. Then 300–800 g of slurry was amended to 1 L DURAN flasks and immediately treated with different acids alone or in combinations. The slurry-treated flasks were then placed into a continuous gas monitoring setup. In the setup wet air was continuously dosed with a mass flow controller (Bronkhorst EL-FLOW, Ruurlo, Netherlands) and distributed

evenly to the headspace of the 15 x 1 L flasks containing treated pig slurry. An empty flask was used for background measurements. The continuous air flow was adjusted to 1.3 L/min per slurry flask. The outlet flow from the flasks was directed to a 16-port manifold (Picarro, Santa Clara, CA, USA), which changed position between the 16 flasks every 15th minute. An exhaust tube was placed before the manifold to avoid pressure buildup when the manifold was closed for a particular flask. The concentration of $CH_4$, $N_2O$, and $CO_2$ from the flasks was measured with a G2508 cavity ring-down spectrometer (Picarro) with extended $CH_4$ concentration range. Due to spectral interference at high concentration levels, $NH_3$ was removed by applying an $H_2SO_4$ scrubber and a Nafion tube before the inlet of the G2508 as previously described [22].

In total three CHS experiments each lasting 2 weeks were carried out. A presentation of the treatments is detailed in Table 3. In CHS experiment A, 800 g of pig slurry (> two months old) with considerable methanogenesis activity (observed from in-situ measurements before sampling), was added to 1 L incubation flasks. Then 10–20 g $HNO_3$ (69% purity) (kg slurry)$^{-1}$ was added to the slurry flasks to reach pH values between 5.5 and 6.25. Acidification in all CHS experiments was carried out by alternately pipetting acid into the slurry under agitation with a magnet stirrer followed by pH measurement. This procedure was repeated until the pH reached and remained stable at the target pH over 5 minutes. In CHS experiment A, nitrous oxide was measured continuously over two weeks, but $CH_4$ and $CO_2$ were only measured periodically when a Nafion filter was applied to remove $NH_3$ and possibly other volatile organic compounds, which caused spectral interference with $CH_4$ and $CO_2$. On the contrary, $NH_3$ was measured in periods where the Nafion filter was not applied. In CHS experiment B, 500 g of

**Table 3. Treatment details of continuous headspace experiments.**

| Treatment | Dose at experiment start (g (kg slurry)$^{-1}$) | Slurry start + daily addition (g) |
|---|---|---|
| CHS experiment A, datetime; 21-10-2020–05-11-2020, Fig 4 | | |
| Untreated | – | 800 |
| Nitric acid to pH 6.25 | 10.11 ± 0.46 | |
| Nitric acid to pH 6.00 | 12.29 ± 0.04 | |
| Nitric acid to pH 5.75 | 15.47 ± 0.10 | |
| Nitric acid to pH 5.5 | 17.51 ± 0.04 | |
| CHS experiment B, datetime; 11-11-2020–25-11-2020, Fig 3 | | |
| Untreated | – | 500 |
| Sulfuric acid to pH 5.5 | 11.29 ± 0.15 | |
| Nitric acid to pH 5.5 | 20.23 ± 0.04 | |
| Nitric acid (L)[a] and sulfuric acid to pH 5.5 | 5.28 ± 0.05 and 7.94 ± 0.06 | |
| Nitric acid (H)[a] and sulfuric acid to pH 5.5 | 10.62 ± 0.13 and 6.29 ± 0.28 | |
| CHS experiment C, datetime: 26-11-2020–17-12-2020, Fig 4 | | |
| Untreated | – | 300 + 78 |
| Sulfuric acid to pH 5.5 | 11.68 ± 0.017 | |
| Nitric acid to pH 5.5 | 19.15 ± 0.28 | |
| Nitric acid (L)[a] and sulfuric acid to pH 5.5 | 5.51 ± 0.07 and 8.63 ± 0.11 | |
| Nitric acid (H)[a] and sulfuric acid to pH 5.5 | 10.44 ± 0.04 and 5.35 ± 0.02 | |

[a] (L) and (H) refers to "low" and "high" doses of nitric acid.

residual pig slurry was added to 1 L incubation flasks. Compared to CHS experiment A, the slurry mass was reduced (from 800 to 500 g) to avoid excessive foam formation during acidification. The slurries were acidified with either $HNO_3$ (69%), $H_2SO_4$ (96%), or mixtures of both acids to final pH values of 5.5 for all treatments. Greenhouse gases were monitored continuously the following two weeks with the Nafion filter permanently applied. For CHS experiment C, the intention was to simulate weekly slurry flushing combined with residual slurry acidification. To start with, 300 g residual slurry was added to 1 L incubation flasks and the slurries were acidified with $HNO_3$ (69%), $H_2SO_4$ (96%), or mixtures of both to final pH values of 5.5 for all treatments. On the day after treatment (day 1) and on a diurnal basis hereafter, 78 g of vacuum flushed slurry (<7 days old) was added to all incubation flasks to simulate daily excretion of slurry from the animals, resulting in a total of 768 g slurry in each flask on day 6. On day 7, the slurries were mixed by gently shaking and slurry was removed until 300 g was left in each flask to simulate frequent slurry removal practice. The remaining slurries were then reacidified (residual acidification) with $HNO_3$ (69%), $H_2SO_4$ (96%), or mixtures of both, to final pH values of 5.5. Later on day 7, 78 g vacuum flushed slurry (<7 days old) from a new slurry aliquot was amended and diurnal slurry addition was repeated until and including day 13. Greenhouse gases were measured continuously and with the Nafion filter permanently applied. For all CHS experiments, the pH was measured every 3–4th day by opening the incubation flasks completely and carefully submerging the pH sensor to 1 cm above the bottom of the incubation flask without stirring the slurry. All treatments in continuous headspace experiments A, B, and C were carried out in triplicates.

## Gas analysis and emission calculations

For batch screening (BS) experiments $CH_4$ concentration in the headspace of the vials was measured on a GC-TCD (Agilent Technologies 7890A, USA) equipped with a Porapak Q column (Agilent) and argon carrier gas. The pressure in the headspace of the vials was measured with an MPX4250AP absolute pressure sensor. $CH_4$ production was calculated as the product of vial headspace pressure, vial headspace volume, and vial headspace $CH_4$ concentration.

In continuous headspace experiments (CHS), a G2508 analyzer (Picarro, Santa Clara, CA, USA) was used to measure CH, $N_2O$, $CO_2$, $NH_3$, and $H_2O$ concentrations. For $NH_3$, humidity correction was done according to [23], using Eq 1.

$$NH_{3,dry} = (NH_{3,raw} + 1.73 \cdot 10^{-8} \cdot H_2O)/(1 + (-1.78 \cdot H_2O + 2.35 \cdot H_2O^2)) \qquad (1)$$

Where $NH_{3,raw}$ is the output from the G2508 analyzer. Gas emission rate was calculated using Eq 2.

$$r_j = C_j \cdot Q \cdot \rho_j \qquad (2)$$

Where $r_j$ is the emission rate (g day$^{-1}$) of gas $j$, $C_j$ is the gas concentration (atmosphere) of gas $j$ measured by the CRDS instrument, $Q$ is the airflow rate (L day$^{-1}$) in the headspace of each incubation flask, and $\rho_j$ is the density of gas $j$ at 25°C. The cumulated gas emission was calculated by the trapezoidal method of integration in Eq 3.

$$E_{j,i} = E_{j,i-1} + \frac{1}{2} \cdot \left(r_{j,i} + r_{j,i-1}\right) \cdot (t_i - t_{i-1}) \qquad (3)$$

Where $E_{j,i}$ is the cumulative gas emission (g) of gas $j$ after $i$ rate measurements, $r_{j,i}$ is the emission rate (g day$^{-1}$) of gas $j$ at $i^{th}$ rate measurement, and $t_i$ is the time (days) at rate measurement $i$.

Emission in $CO_2$ equivalents ($CO_2$-eq) was calculated using 100-year global warming potentials without inclusion of climate-carbon feedbacks of 273 ($N_2O$) and 27.2 ($CH_4$) [14]. Carbon dioxide emission was not included in the calculation of $CO_2$-eq as the $CO_2$ flux to and from the atmosphere is considered to be net-zero from livestock animals [3].

The non-dissociated form of acetate (acetic acid or HAc) was calculated from Eq 4, using a pKa value for HAc of 4.76

$$\frac{HAc}{Ac + HAc} = \frac{1}{1 + 10^{(pH - pka)}} \qquad (4)$$

## Statistics

Error bars in figures and tables represent sample standard deviations. For batch experiments with many different treatments, statistical significance and reported p-values were obtained from one-way ANOVA tests using a level of significance ($\alpha$) of 0.05. The ANOVA test was done in MatLab using the "anova1" function with default settings. The statistics from "anova1" were used as input for a pairwise comparison between means of treatments to indicate which treatments differed from each other. The pairwise comparison was done using the "multcompare" function in MatLab with the default Tukey Kramer post hoc method. For continuous headspace experiments, two-sample t-test with assumed unequal variance was conducted to test for the difference of individual treatments compared to the untreated slurry. The level of significance ($\alpha$) was 0.05 for the t-tests, which was conducted in MS Excel 2016. The abbreviation "ns" is applied when presented results that are not statistically significant in the text. Error propagation was used for calculating errors on $CO_2$-eq and where otherwise relevant according to [24]. For rate plots (Figs 2A, 3A, and 4A) data is presented as rolling means over 12 h using the "zoo" R-package.

## Results

### Batch screening experiments

The batch experiments consisted of three separate experiments, where the approximate effect of organic, inorganic, and mixtures from both groups was assessed with respect to their inhibiting effect on $CH_4$ production. Fig 5 shows inhibition potential by presenting the cumulative $CH_4$ production in vials incubated with acid-treated pig slurries (experimental details in Table 2). Both organic and inorganic acids all caused a significant inhibition of $CH_4$ production compared to untreated slurry (Fig 5A and 5B). Nitric acid was the most effective inhibitor with > 99% $CH_4$ reduction at pH 5.5 (Fig 5A and 5B). The inhibition effect of acid treatments lasted throughout the experimental period with only untreated slurry producing $CH_4$ from day 6 to day 18. The untreated slurry had an initial acetate (HAc + Ac) concentration of 7.26 ± 0.95 g/L and there was no effect of combining acetic acid with the inorganic acids (Fig 5B) as compared to inorganic acids alone when acidifying to a final pH of 5.5. Different organic acids and concentrations thereof were tested to examine potential synergistic mitigation effects with $H_2SO_4$ to hereby reduce the total volume of acids needed to inhibit $CH_4$ production. However, no synergistic effects were observed (Fig 5C). Most of the acid treatments resulted in a pH increase from 5.5 to pH 5.6–6.15 by the end of the experiments, and the pH increase was positively correlated with the amount of $CH_4$ produced (S1 Appendix). Formic acid + $H_2SO_4$ increased pH to 6.48–6.79 and stimulated $CH_4$ production compared to untreated slurry (Fig 5C). Similarly, lactic acid alone increased pH to 7, but $CH_4$ was still reduced compared to the untreated slurry in Fig 5A. A smaller dosage of lactic acid in combination with $H_2SO_4$ produced equal amounts of $CH_4$ compared to the untreated slurry in Fig 5C.

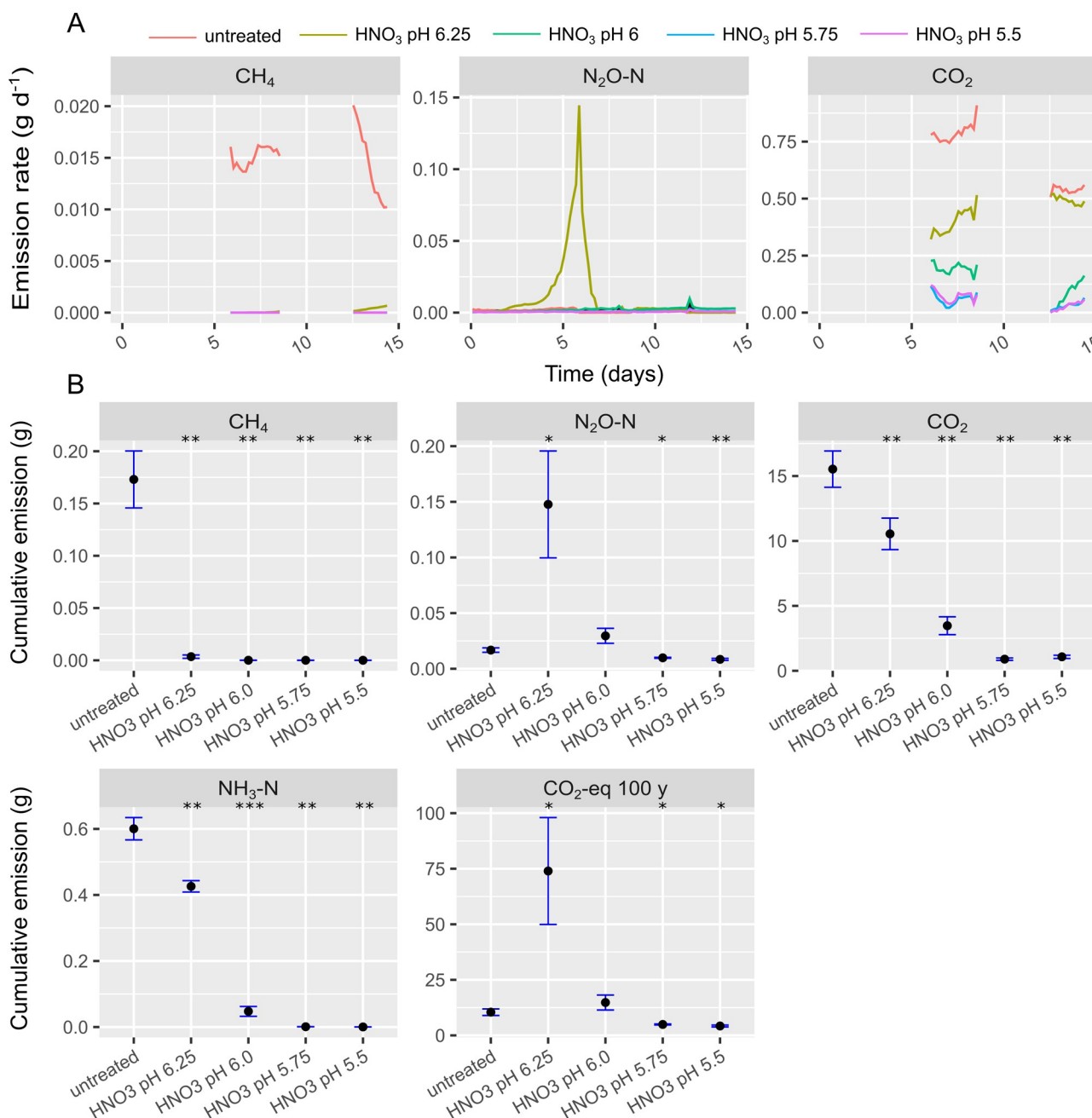

**Fig 2.** Gas emission rates (A) and cumulative gas emissions (B) from continuous headspace experiment A. In (A) colors indicate different treatments according to legend. In (B) black dots represent average of triplicates and the blue error bars indicate sample standard deviation. Treatments that were statistically different from the untreated slurry concerning cumulative emission are indicated with * ($p < 0.05$), ** ($p < 0.01$), *** ($p < 0.001$).

### Continuous headspace experiments

The batch screening experiments suggested that $HNO_3$ was an efficient $CH_4$ inhibitor and in the subsequent continuous headspace experiments, the effect of $HNO_3$ was studied in more detail.

Fig 6 presents pH profiles from the three CHS experiments. In CHS experiment A, pH values were relatively stable, except for the $HNO_3$ to pH 6.25 treated slurry, which increased from

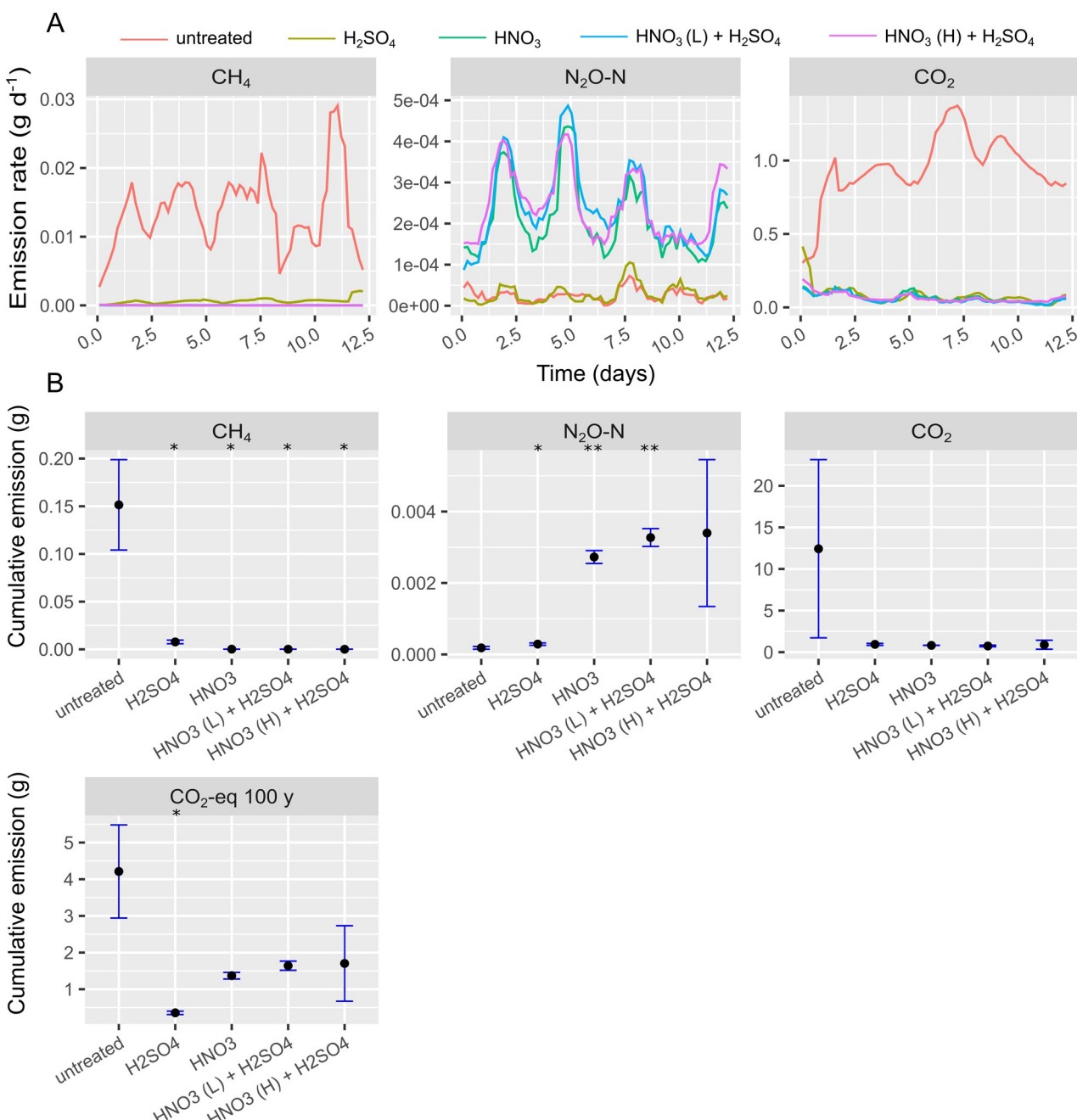

**Fig 3.** Gas emission rates (A) and cumulative gas emissions (B) from continuous headspace experiment B. In (A) colors indicate different treatments according to legend. In (B) black dots represent average of triplicates and the blue error bars indicate sample standard deviation. Treatments that were statistically different from the untreated slurry concerning cumulative emission are indicated with * ($p < 0.05$), ** ($p < 0.01$), *** ($p < 0.001$).

pH 6.25 to 7.25 during the first week. The reason that the variant with $HNO_3$ to pH 6.25 had a higher end pH than the untreated slurry is unclear, but it is likely a combination of denitrification producing strong base [11] and initial loss of acidic $CO_2$ during acidification. Treatments with $HNO_3$ to pH 6 or below did not change considerably in pH. In CHS experiment B (Fig 6 middle), all treatments to pH 5.5 increased only to pH ~5.75 and stabilized after 2–3 days, independent of the acidic agent used. However, during CHS experiment C, new slurry was

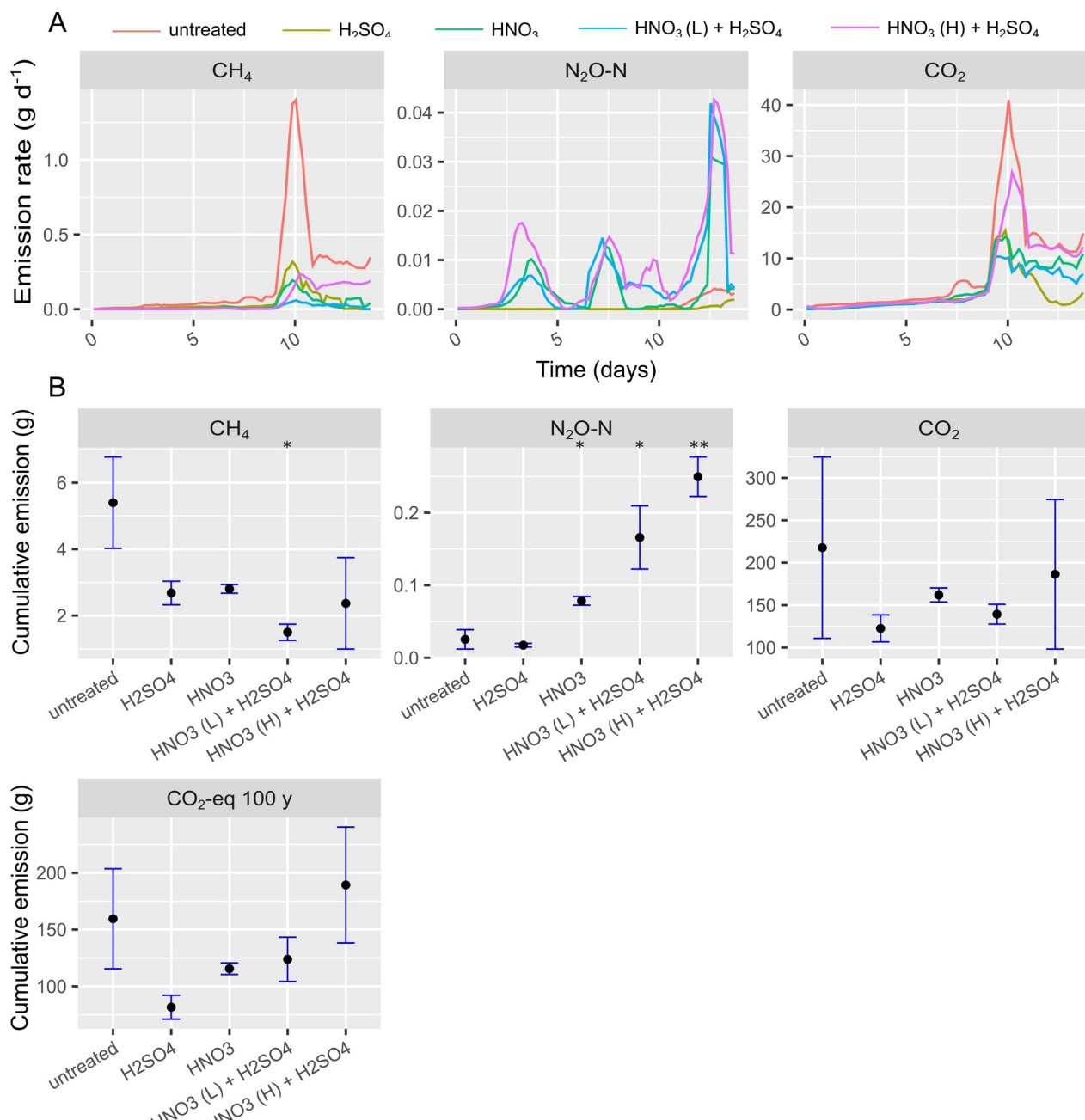

**Fig 4.** Gas emission rates (A) and cumulative gas emissions (B) from continuous headspace experiment C. In (A) colors indicate different treatments according to legend. In (B) black dots represent average of triplicates and the blue error bars indicate sample standard deviation. Treatments that were statistically different from the untreated slurry concerning cumulative emission are indicated with * (p<0.05), ** (p<0.01), *** (p<0.001).

added daily, resulting in a significant pH increase of more than 1 pH unit during week one and again during week two.

Fig 2 shows gas emission rates (a) and cumulative gas emission (b) from CHS experiment A, where slurry was acidified to different pH values using $HNO_3$. Spectral interference from high $NH_3$ levels in periods without a Nafion filter applied resulted in erroneous $CH_4$ measurements and has been omitted. Nitric acid reduced $CH_4$ emission severely, but treatment with $HNO_3$ to pH 6.25, which was the lowest dosage of $HNO_3$, showed a trend of increasing $CH_4$

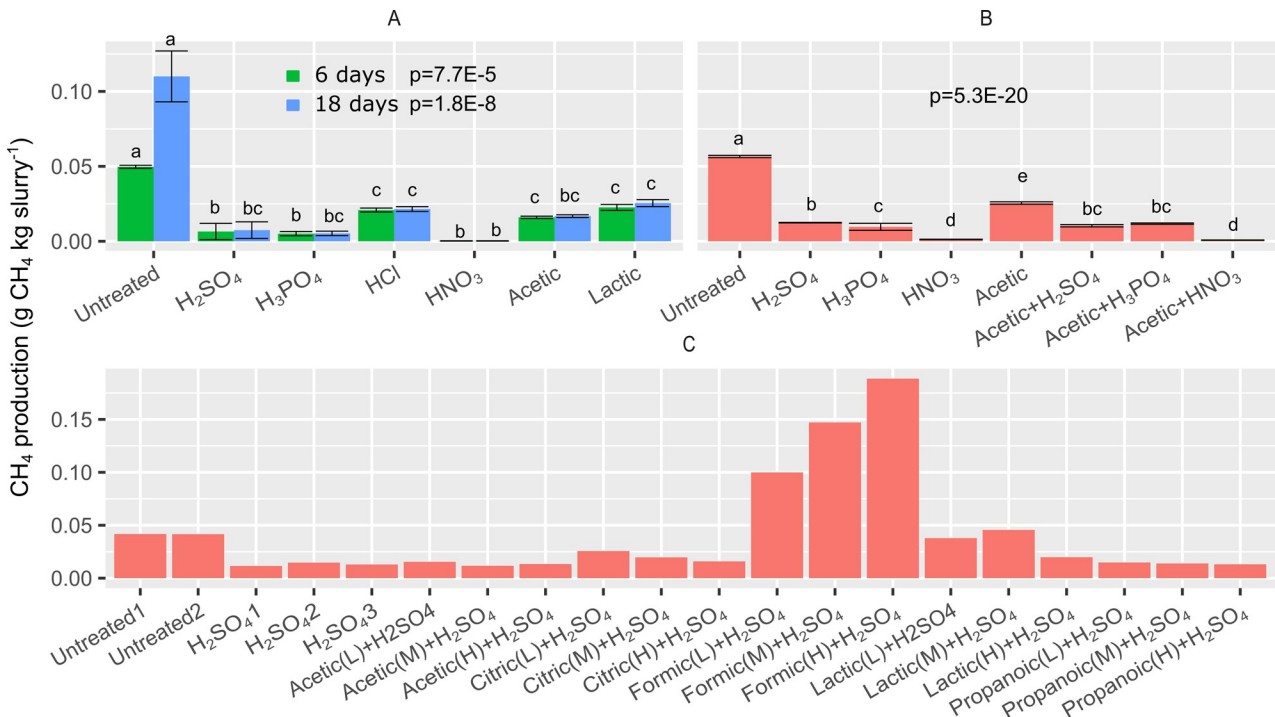

**Fig 5.** Inhibition potential of different acids on pig slurry in batch screening experiments A (A), B (B), and C (C). Uncertainty bars indicate standard deviation. In (C), (L), (M), and (H) indicate low, medium, and high dose treatments as described in Table 2. *p*-values are for one-way ANOVA test on differences between $CH_4$ production and bars with letters in common are not statistically different with a pairwise comparison of means (Tukey Kramer).

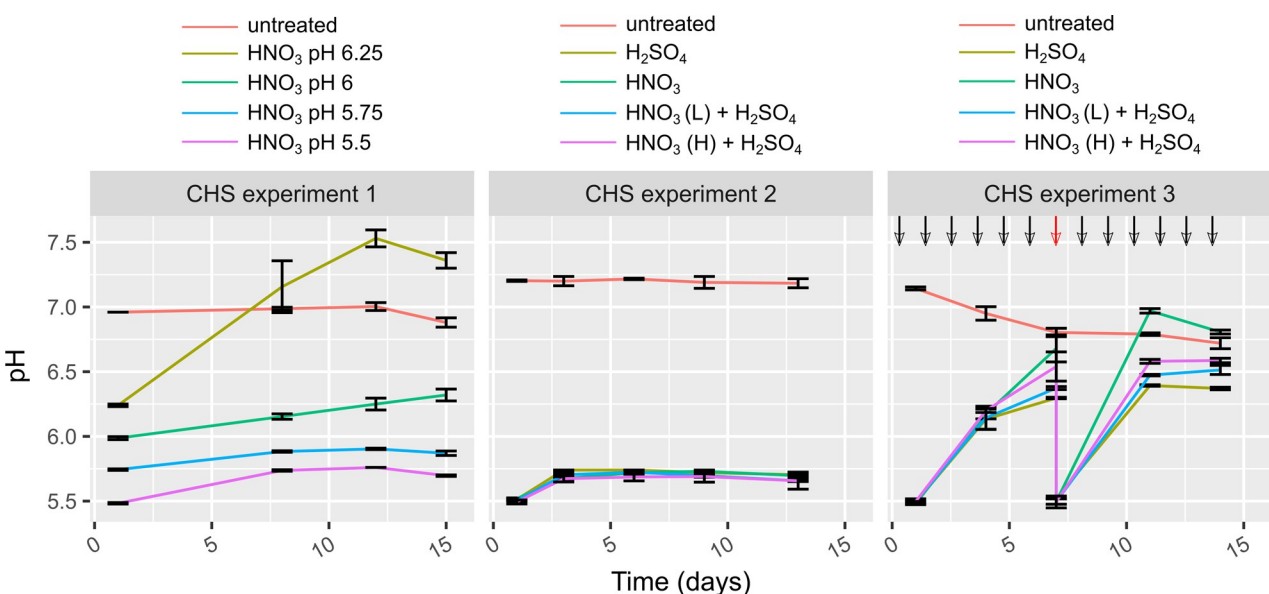

**Fig 6.** Mean pH values measured in continuous headspace experiments A, B, and C. Black arrows in CHS experiment C indicate addition of vacuum flushed slurry and the red arrow indicates addition of slurry + slurry removal + acidification. Uncertainty bars indicate sample standard deviation of triplicates.

emission by the end of the experiment (Fig 2A). Slurry treatment with $HNO_3$ to pH 6.25 produced a considerable $N_2O$ emission peak around day 5. The simultaneous pH increase (Fig 6), suggests that $N_2O$ was produced from denitrification of $NO_3^-$ from $HNO_3$. Ammonia emission from slurry increased with increasing pH due to a shift from nonvolatile $NH_4^+$ to $NH_3$ (pka = 9.25) (S2 Appendix). The significantly higher $N_2O$ emission from slurry treated with $HNO_3$ to pH 6.25 (p = 0.042) resulted in significantly higher emission of $CO_2$-eq (p = 0.044) than the untreated slurry. However, with $HNO_3$ treatment to pH 5.75 and pH 5.5, $CO_2$-eq emission was significantly reduced by 52.9% (p = 0.021) and 59.5% (p = 0.013) compared to untreated slurry.

As $N_2O$ emission was very pronounced at pH 6.25, another round of experiments was conducted in which all slurries were acidified to pH 5.5 and the N-pool available for microbial production of $N_2O$ was minimized by replacing different amounts of $HNO_3$ with $H_2SO_4$ (Table 3). Fig 3 shows gas emission rates from CHS experiment B, with slurry acidified with mixtures of $HNO_3$ and $H_2SO_4$ to pH 5.5. All treatments with $HNO_3$ reduced $CH_4$ emission rates by approximately three orders of magnitude (> 99.9%) compared to the untreated pig slurry and more than two orders of magnitude (> 99%) compared to $H_2SO_4$ treatment (Fig 3). These reductions are difficult to perceive from Fig 3, as all treatments showed close to zero $CH_4$ emission and we refer to S3 Appendix for log10 transformed y-axis. The $N_2O$ emission from treatments with $HNO_3$ was 14.8–18.4 fold higher than from untreated slurry (p = 0.001–0.1), indicating that denitrification still proceeded at pH 5.5, however at very limited rates compared to CHS experiment A at pH 6.25. Carbon dioxide emission was lower for all acidification treatments compared to untreated slurry (ns, p = 0.2) suggesting that microbial oxidation of substrates and intermediates, during either hydrolysis, fermentation, or methanogenesis, was severely slowed down at pH 5.5. In terms of $CO_2$-eq, untreated slurry had higher emission than acidification treatments, primarily due to the higher $CH_4$ emission. The most effective inhibitor was $H_2SO_4$ with 91.5% $CO_2$-eq emission reduction (p = 0.034). Reduction of $CO_2$-eq emission with $HNO_3$ treatments was 67%–59% (ns, p = 0.059–0.071).

To simulate frequent slurry removal and daily excretion of pig feces and urine, treatments from CHS experiment B were repeated, but with daily addition of slurry and weekly slurry removal to a threshold level, which was equivalent to a situation with weekly vacuum flushing in a pig pen. In addition, the residual slurry was acidified after one week to reduce acid consumption and target the methanogenic inoculum. During the first week in CHS experiment C, $CH_4$ emission was lower for all $HNO_3$ treatments, but during week two, $CH_4$ emissions accelerated to levels nearly identical to those from untreated slurry (Fig 4). This resulted in total $CH_4$ reductions ranging from 48–72% compared to the untreated slurry, with $HNO_3$ (L) + $H_2SO_4$ being the most effective treatment based on cumulative emission and the only treatment that was significantly different from the untreated slurry (p = 0.036, others p = 0.054–0.081) (Fig 4B). Nitrous oxide emission showed a less dramatic increase during week two than $CH_4$ and the increase was more pronounced for slurries with $HNO_3$ addition, which is consistent with results from CHS experiment B. The $CO_2$-eq emission was lowest for $H_2SO_4$, with a reduction of 49% compared to untreated slurry (ns, p = 0.085) and 27.5% reduction for $HNO_3$ treatment (ns, p = 0.225). Based on these observations pure $H_2SO_4$ acidification of residual slurry would be preferable from a 100-year climate perspective. This was mainly due to the $N_2O$ production outweighing the extra reduction of $CH_4$ gained by using $HNO_3$ instead of $H_2SO_4$.

## Discussion

### Batch screening experiments

The batch screening of combined acidic agents was initiated with expectations that synergistic effects could occur due to the uncoupling effect previously described [9]. This theory has not

been properly tested in stored animal slurry until recently, where Fuchs et al (2021) reported that $CH_4$ emission from acidified cattle slurry showed a larger reduction with acetic acid treatment than $H_2SO_4$ treatment (both to pH 5.5). Fuchs et al. (2021) reported 97% $CH_4$ inhibition at 1631 mg HAc/L in cattle slurry acidified to pH 5.5 at room temperature. In comparison, $H_2SO_4$ and acetate reduced $CH_4$ by 93% and 85%, respectively over 18 days of incubation (Fig 5A) in the experiments reported here. Zhang et al. 2018 also studied methanogen inhibition in the pH range of 5–7 and acetate concentration ranging from 0–15 g/L and found that methanogen activity could not be recovered when exceeding 810 mg HAc/L [25]. In this study, we did not observe additional $CH_4$ reduction with acetate addition. The acetate concentration in the untreated slurry was 7.26 ± 0.95 g/L, which equals 1119 ± 147 mg HAc/L at pH 5.5 (calculated from Eq 4). This being well above the recovery limit of 810 mg HAc/L reported by Zhang et al. (2018), we deem that further addition of acetate would have no additional inhibitory effect on methanogenesis. Importantly, this finding is not conflicting with previous literature findings, but highlights that utilization of acetate or other organic acids (Fig 5C) as combination inhibitors with $H_2SO_4$, will probably only be beneficial in a relatively VFA depleted slurry similar to that used by Fuchs et al. (2021), which contained 104 ± 12 mg HAc/L at pH 5.5 ($H_2SO_4$ acidified). Acetate treatment alone resulted in a final pH of 6.05 and a larger $CH_4$ production (Fig 5B) than other treatments (all pH 5.5), although it still reduced production by 55% compared to the control. The lesser inhibition was likely due to a smaller proportion of the acetate (and other organic acids) being protonated at pH 6.05 than at pH 5.5. But the absence of sulfate or reduced sulfur species from $H_2SO_4$, which have earlier been reported to inhibit methanogenesis [7], could also explain lesser inhibition. The latter theory of sulfur species hampering methanogenesis is consistent with results in Fig 5A since $H_2SO_4$ acidification (pH 5.5) was more efficient than pure acetate acidification (pH 5.5) and HCl acidification (pH 5.5).

CH_4 production was observed for acidified pig slurry with formic acid addition (Fig 5C) and it is well known that formate is an electron donor of $CH_4$ production [26]. Still, at an initial pH of 5.5, formic acid fueled $CH_4$ production and exceeded $CH_4$ production in untreated slurry. This suggests that both pH and acid type are important predictors of $CH_4$ inhibition potential in acidified slurry. This finding disqualifies formic acid as an inhibitory agent even under acidic conditions. The high $CH_4$ reduction potential reported by Berg et al. (2006) for lactic acid was not confirmed in our screenings experiments. Instead, an increase in pH (S1 Appendix) and limited effect on $CH_4$ production was observed with lactic acid treatments. This discrepancy could be related to differences in acidification target pH (pH 3.8–4.8 versus 5.5) and slurry type (cattle versus pig), which would affect buffer capacity and microbial community. Importantly buffer systems in slurry can result in pH increases after acidification. Hence, the $CH_4$ reductions reported from the batch screening experiments are not comparable to commercial in-house acidification systems with continuous pH adjustment but instead hint to the relative efficacy of the different acids in such a system.

### Continuous headspace experiments

In CHS experiment A, a clear tendency of increasing pH was observed for $HNO_3$ acidification to pH 6.25 (Fig 6). This pH increase could be related to release of acidic $CO_2$ (Fig 2A) produced from organic matter transformation by microbes. For example, dissimilatory nitrate reduction to ammonium or denitrification of $NO_3^-$ with acetate as electron donor produce strong base thereby increasing pH [11]. The $N_2O$ peak in Fig 2A suggests that denitrification of $NO_3^-$ indeed occurred during the period where the pH increased for the $HNO_3$ treatment to pH 6.25. Earlier experimental work has shown that denitrification produced high $N_2O$ fluxes

from stored cattle slurry acidified with $HNO_3$ to pH 6.0 but at pH < 5 $N_2O$ flux was reduced by several orders of magnitude [16]. That trend was demonstrated again in the current study, where $N_2O$ emission was completely inhibited at pH < 6 (reduced > 99% compared to $HNO_3$ to pH 6.25). The optimum pH for denitrification in soils is neutral to slightly alkaline [27], which is in line with an interpolated pH value around ~7 at day 5–6 when the $N_2O$ peak is at its highest (Figs 2A and 6). A mass balance revealed that $N_2O$ emission corresponded to ~ 12% of the added $NO_3^-$–N from $HNO_3$, suggesting that the remaining $NO_3^-$ was either not reduced or reduced through complete denitrification to $N_2$ or dissimilatory $NO_3^-$ reduction to $NH_4^+$. For slurries with pH ≤ 6 (Fig 6) there was only a small pH increment in the beginning and hereon after a pH stabilization 0.1–0.25 pH units above the initial pH. The small pH increase could be due to release of retained $CO_2$, volatile fatty acids, or hydrogen sulfide produced during the acidification process. It has been shown that enzymatic $N_2O$ reductase activity is significantly reduced at low pH [28], thereby decreasing the enzymatic reduction of $N_2O$ to $N_2$. However, the fact that $CO_2$ production was reduced below pH 6, suggests that general inhibition of heterotrophic activity (including heterotrophic denitrifiers) is a more plausible explanation for the reduced $N_2O$ emission below pH 6. Given these considerations, a critical threshold for microbial activity likely exists somewhere between pH 6 and pH 6.25. This is consistent with the batch experiment, where acetate alone to pH 6.05 produced $CH_4$ to a larger extent than treatments to pH 5.5, but still not as much as in untreated slurry (Fig 5B).

Emission of $N_2O$ was diminished substantially in CHS experiment B for $HNO_3$ treatments, but still, they were significantly higher than the untreated slurry, probably due to a limited but persistent denitrification activity (Fig 3). Conversely, $CH_4$ was reduced more in $HNO_3$ treatments, which we attribute to increased competition for substrates by denitrifiers in the presence of $NO_3^-$. To assess the most efficient acidic agent a reasonable approach is a comparison of the total climate impact in terms of $CO_2$-eq. Here we did not find evidence that suggests $HNO_3$ is a superior greenhouse gas inhibitor to $H_2SO_4$. The CHS experiment A and B were conducted without daily slurry addition and without optimizing the timing of acidification. This justified carrying out the final CHS experiment C, where new slurry was added daily. Here the initial low pH of acidified slurries was expected to increase (Fig 6) as the vacuum flushed slurry had a pH of 6.7–6.8 (see Table 1). Addition of new slurry would dilute and thereby neutralize the acidic environment and add new substrates and microbial biomass to fuel organic matter transformation. This pH increase resulted in increasing $CH_4$ production rates, which accelerated quickly at day 9 (Fig 4). This indicates that residual acidification by day 7, was overcome more quickly than at day 0, most likely due to a more adapted microbial community in week two. Considering the development in emission patterns during the two weeks of monitoring, the $CH_4$ reduction potential is likely to decrease with time and this should be assessed in long-term experiments with more cycles of flushing and acidification. In comparison to the 48–72% $CH_4$ reduction in CHS experiment C, Petersen et al. (2012) reported a 67–87% reduction in $CH_4$ emission from $H_2SO_4$ bulk acidified cattle slurry in pilot storage tanks, without addition of new slurry. Sokolov et al. (2020a; b), achieved high $CH_4$ reductions (>99% in lab; 77% in pilot storages) with residual acidification, but similarly to other studies, new slurry was not amended. Therefore, the abovementioned studies are perhaps more comparable to CHS experiment B without slurry addition and slurry removal where we observed > 99% $CH_4$ reduction. Only a few studies have studied the effect of acidification (bulk or residual) on greenhouse gas emission from pig barns, where slurry is naturally and continuously added. Petersen et al. 2016 measured 50% $CH_4$ reduction (not significant) from a pig barn in a spring campaign, but low $CH_4$ emissions from the slurry compared to enteric emission reduced the confidence of the evaluation [29]. A report by the Danish knowledge institution, SEGES, reports a 60% methane reduction from bulk slurry acidification

including enteric $CH_4$ emission [30]. The mentioned studies both utilized photo-acoustic spectroscopy for $CH_4$ monitoring, which suffers from spectral interference of volatile organic compounds and $H_2O$ [31, 32]. Based on results from the available literature Olesen et al. 2018 assessed that a $CH_4$ reduction of 60% for bulk acidification in barns, including enteric $CH_4$ emission, is realistic [33]. Assuming that 30% of $CH_4$ comes from enteric fermentation, $CH_4$ reductions from the slurry is probably > 90%, which as expected is higher than in CHS experiment C where only residual acidification was tested. Notably, there was not a significant effect of any acidification treatment on the $CO_2$-eq emission in CHS experiment C, which is a consequence of the chosen t-test with unequal variances, which is more conservative. Using the regular student's t-test with assumed equal variance would increase statistical power and result in a statistical difference for $H_2SO_4$ treatment compared to the untreated slurry, but the chance of concluding a Type I error would also increase [34]. In summary, CHS experiment C shows that a complete evaluation of residual slurry acidification with weekly slurry removal in pig barns would require more cycles of flushing and should be conducted using $H_2SO_4$. The reduced effect of residual slurry acidification compared to bulk acidification suggest that residual slurry acidification should be carried out with a lower target pH than 5.5 to maintain the inhibiting effect once new slurry is added. It is necessary to examine if the residual slurry can form an adapted microbial community, which will reduce the long-term effects of methanogenic inhibition through acidification.

## Perspectives of acidification strategy

The described CHS experiments as well as the initial BS experiments in general find no compelling reason to shift to other acidic agents than $H_2SO_4$ when taking into account greenhouse gas emission from the slurry alone. From a farm-scale perspective, in certain scenarios, supplementing or acidifying slurry with other organic acids could be beneficial due to increased nutritional value as field fertilizer [17], and the risk associated with handling hazardous $H_2SO_4$ would be eliminated [35]. In addition, if the slurry is further processed in anaerobic digesters to produce biogas, acetic acid could hold potential as a methanogen substrate under controlled feeding. On the contrary, $H_2SO_4$ acidification, hampers downstream use of the slurry in biogas reactors, unless it is co-digested with untreated slurry at a maximum of 10% acidified slurry [36]. However, the degradation rate of organic acids is rapid compared to $H_2SO_4$ [37] and the timing of pure organic acid treatment is therefore critical in this context. In slurries with limited nitrogen, $HNO_3$ may be supplied to increase the nutritional fertilizer value when landspreading the slurry to crops fields, but comes with a considerable risk of increasing $N_2O$ emissions [37].

## Conclusion

Two types of experiments were conducted; 1) batch vials experiments and 2) continuous headspace experiments. In the batch vial experiments, $CH_4$ production was reduced by >99% with $HNO_3$ treatment to pH 5.5. Sulfuric acid and other inorganic and organic acids also reduced $CH_4$ production significantly, but no synergistic effects between mixed acids were observed. It is likely that high concentrations of VFAs present before acidification reduce the efficacy of organic acid treatment. In the continuous headspace experiments, three rounds of experiments were conducted. Nitric acid treatment to pH 5.5, 5.75, 6.0, and 6.25, revealed that the degree of inhibition increased with lower pH for both $CH_4$ and $N_2O$ emissions. Nitric acid treatment to pH 6.25 produced significant amounts of $N_2O$ and a concurrent pH increase suggested that this was due to denitrification activity. For bulk acidification with $H_2SO_4$ to pH 5.5 a 91.5% reduction in $CO_2$-eq was found, whereas $HNO_3$ and mixtures of $HNO_3$ with $H_2SO_4$ reduced

$CO_2$-eq by only 59–67% due to increases in $N_2O$ emission. When simulating weekly removal of slurry combined with acidification of the residual slurry and daily addition of fresh slurry (CHS exp C), a considerable increase in greenhouse gas emissions was observed for all treatments during week two, resulting in a $CO_2$-eq reduction of 48% for $H_2SO_4$ (ns) and 27% for $HNO_3$ (ns). Further in CHS exp C, $CH_4$ was reduced significantly only for $HNO_3$ mixed with $H_2SO_4$ by 72%, but $N_2O$ emission largely outweighed this reduction. This suggests that when slurry is added daily and the microbial community adapts, acidification of the residual slurry becomes less efficient, but further studies are necessary to determine more clearly the long-term effect of adaptation mechanisms related to residual slurry acidification.

## Supporting information

**S1 Appendix. Correlation between $CH_4$ and pH.**
(DOCX)

**S2 Appendix. $NH_3$ emission rate.**
(DOCX)

**S3 Appendix. log10 transformed emission rates.**
(DOCX)

**S4 Appendix. Data for figures.**
(XLSX)

## Acknowledgments

We thank Robin Due Nielsen for carrying out laboratory experiments related to this work.

## Author Contributions

**Conceptualization:** Frederik R. Dalby, Michael V. W. Kofoed.

**Data curation:** Frederik R. Dalby.

**Formal analysis:** Frederik R. Dalby.

**Funding acquisition:** Anders Feilberg.

**Investigation:** Frederik R. Dalby, Michael V. W. Kofoed.

**Methodology:** Frederik R. Dalby, Lise B. Guldberg, Anders Feilberg, Michael V. W. Kofoed.

**Project administration:** Anders Feilberg, Michael V. W. Kofoed.

**Resources:** Lise B. Guldberg, Anders Feilberg, Michael V. W. Kofoed.

**Software:** Frederik R. Dalby.

**Supervision:** Frederik R. Dalby, Anders Feilberg, Michael V. W. Kofoed.

**Validation:** Frederik R. Dalby.

**Visualization:** Frederik R. Dalby.

**Writing – original draft:** Frederik R. Dalby, Lise B. Guldberg.

**Writing – review & editing:** Frederik R. Dalby, Lise B. Guldberg, Anders Feilberg, Michael V. W. Kofoed.

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
