## [Decision Letter · Decision Letter 0]

9 Feb 2022

PONE-D-22-02024Reducing greenhouse gas emissions from pig slurry by acidification with organic and inorganic acidsPLOS ONE

Dear Dr. Dalby,

Thank you for submitting your manuscript to PLOS ONE. After careful consideration, we feel that it has merit but does not fully meet PLOS ONE’s publication criteria as it currently stands. Therefore, we invite you to submit a revised version of the manuscript that addresses the points raised during the review process.

We look forward to receiving your revised manuscript.

Kind regards,

Xueming Chen

Academic Editor

PLOS ONE

" No"

"The research was funding by the Danish Agricultural Agency under the Ministry of Environment and Food (Miljø- og Fødevareministeriet) (grant no. 33010-NIFA-19-725) and would like to thank for the opportunity to conduct this research. "

"NO"

**Comments to the Author**

Reviewer #1: OVERVIEW AND GENERAL RECOMMENDATION

1. The study gives a comprehensive overview of the use of different acids for acidification on greenhouse gas emissions from slurry. The continuous addition of slurry, as is also common in barns, was also investigated. This leads to a special character of the manuscript. However, the studies are very extensive and are divided into several sub-studies. It is therefore difficult for the reader to follow the intentions of the authors.

2. In my opinion, the title is not appropriate because it mentions organic acids. However, most of the study deals with the inorganic acids H2SO4 and HNO3. Even in the abstract, acetic acid is only briefly mentioned once. Additionally, the authors should point out to the reader which of the acids mentioned are inorganic or organic acids.

3. Why was the pH value of 5.5 in BS chosen?

4. There are buffer systems in the slurry which lead to an increase in the pH value after acidification. This was not taken into account or mentioned. How practically relevant are the studies from BS in which acidification was only carried out once?

5. The figures look blurred. Please improve the resolution.

6. The actual data from which the mean values were formed were not provided.

7. It is recommended that the manuscript is read by a native speaker before submission, as there are some linguistic and grammatical errors in the manuscript.

For example

Page 21, l. 407, have instead of has

Page 23, l. 455, was instead of were

Page 23, l. 464 ...and I would like...

DETAILED COMMENTS

1. Page 1, l. 3, Why does it say "Introduction" after the title?

2. Page 3, ll.47-56, This information does not add much to the understanding of the study.

3. Page 4, l. 78, There is a missing space before the source [20].

4. Page 5, l. 95, With state-of-the-art, one space is too many.

5. Page 5, ll.97-98, It is unusual for results to already be presented in the aim in the introduction.

6. Page 6, ll. 120-124, Why were different types of slurry used (especially for CHS experiment 1 and 2)?

7. Page. 7 l. 130, Table 1 is not mentioned in the text.

8. Page 7, l. 130, Why are the abbreviations for the slurry relevant? Are they even mentioned again in the manuscript?

9. Page 7, l. 140, Table 2 instead of Table 1, please check all cross-references again!

10. Page 8, l. 147, BS experiment 1: length 6 or 18 days; write out abbreviations such as H2SO4 or HCl in the text; explain mM in the table heading; BS experiment 3: H2SO4 better mentioned in 2nd place

11. Page 9; ll. 161-163, Show H2SO4 scrubber and Nafion tube in Figure 1 for better understanding.

12. Page 10, l. 165, Table 3 instead of Table 2

13. Page 10, l. 167, The correct spelling is 20 g HNO3 (69% purity) (kg slurry)-1 (please check in the whole manuscript).

14. Page 10, l. 168, Methane is also a greenhouse gas.

15. Page 10, l. 185-187, How was the pH measurement carried out? Was the vessel opened so that air could enter? Was the sample stirred, which also led to oxygen entering and thus influenced the pH development (stronger pH increase as mentioned in other studies)?

16. Page 11, l. 188, the letters "L", "H" and "B" were not explained

17. Page 11, l. 188, Why were different amounts of slurry used in experiment 1 and 2?

18. Page 11, Chapter 2.4 it is better to mention this chapter earlier in the material and methods section, as results have already been presented in Table 1. Alternatively, the results of the slurry characterisation could also be presented in the results chapter.

19. Page 12, l. 207, The abbreviation ATM was not explained

20. Page 14. l. 251, Why not experiment A, B and C instead of 1, 2 and 3. A better naming of the individual sub-experiments would help the reader to understand the manuscript.

21. Page 15, ll. 259-260, It is better to mention this in the material and methods section

22. Page 15, l. 261-262, It is better to mention this in the material and methods section

23. Page 15, l. 265, Why is the pH value of the variant acidified with HNO3 higher than that of the control?

24. Page, 16, l. 293, one point too many at the end of the sentence

25. Page 23, ll. 458-461, Although the authors correctly point out that a daily addition of fresh slurry makes acidification less efficient, this cannot be avoided in practice in the barn. Or do the authors have any ideas?

26. Fig. 2C, Are there any statistics for this illustration?

27. Fig. 3, The pH value normally increases more quickly after acidification of slurry. Presumably, the slower increase in experiment 1 can be explained by the old slurry > 2 months. This slurry is therefore not representative, especially if you think of an in-house slurry acidification.

28. Fig. 4A, Why do the CH4 emissions decrease on day 7/8 and increase again on day 12/13 after the interruption of the measurement period? The CO2 concentration is also lower between days 6-8? Could the H2SO4 scrubber have had an influence?

Reviewer #2: This study studied GHG emissions from acidified pig slurry with different organic and inorganic acids. Results showed HNO3 could result in N2O emissions and therefore not suitable. This study design followed a straightforward while rigorous manner by applying firstly a large amount of screening tests, followed by continuous headspace experiments. I find the results interesting. I have only some minor comments.

1. It was believed the N2O was due to denitrification of NO3-, was this N2O production/accumulation due to acidic condition or inherent partial denitrification?

2. There is no synergy between inorganic and organic acid. Is there synergy between H+ and NO3-/SO4- (as electron acceptors)?

3. You can also discuss the economic implications of these investigated acids.

Reviewer #3: This study investigated the effect of pig slurry acidification with a range of organic and inorganic acids with respect to their CH4 inhibitor potential in a number of batch experiments. There are some issues with this manuscript which need to be addressed before it can be accepted to publish, and they are listed below.

1. Line 97, the subscript should be corrected. Please check similar problems in the text.

2. In Table 1, could the difference in original TAN between the slurries affect the batch experiments?

3. Line 293, a redundant full stop.

4. Line 339, 344 and 348. I don’t know if the references format meet the requirements of PLOS ONE?

5. Line 386-387, have the denitrification activity been measured? If not, it is not appropriate to state like that.

6. Line 446-447, “It is likely that high concentrations of VFAs present prior to acidification reduce the efficacy of organic acid treatment.” The evidence to support the conclusion should be clarified in the text.

---

## [Author Response · Author response to Decision Letter 0]

10 Apr 2022

In the text below we have addressed all comments from the editor and the reviewers. 

This has been checked and corrected

We don’t see any section called “Financial disclosure” in the online resubmission form. If this is the same as the “Financial statement” we would like it to be: 

“This work was funded by the Danish Agricultural Agency under the Ministry of Environment and Food, Denmark (grant no. 33010-NIFA-19-725). The funder provided support in the form of salaries for authors Frederik R. Dalby, Lise B. Guldberg, Anders Feilberg, and Michael V.W. Kofoed but did not have any additional role in the study design, data collection and analysis, decision to publish, or preparation of the manuscript. The specific roles of these authors are articulated in the ‘author contributions’ section.”

" No"

We have included the statement in the cover letter

"The research was funding by the Danish Agricultural Agency under the Ministry of Environment and Food (Miljø- og Fødevareministeriet) (grant no. 33010-NIFA-19-725) and would like to thank for the opportunity to conduct this research. "

"NO"

We have removed the funding information from the acknowledgments section and the manuscript in general. We don’t see any section called “Funding Statement” in the online submission form. 

We would like the following statement to be used, which is also included in the cover letter. 

“This work was funded by the Danish Agricultural Agency under the Ministry of Environment and Food, Denmark (grant no. 33010-NIFA-19-725). The funder provided support in the form of salaries for authors Frederik R. Dalby, Lise B. Guldberg, Anders Feilberg, and Michael V.W. Kofoed but did not have any additional role in the study design, data collection and analysis, decision to publish, or preparation of the manuscript. The specific roles of these authors are articulated in the ‘author contributions’ section.”

 

Comments to the Author

Reviewer #1: OVERVIEW AND GENERAL RECOMMENDATION

1. The study gives a comprehensive overview of the use of different acids for acidification on greenhouse gas emissions from slurry. The continuous addition of slurry, as is also common in barns, was also investigated. This leads to a special character of the manuscript. However, the studies are very extensive and are divided into several sub-studies. It is therefore difficult for the reader to follow the intentions of the authors.

Thanks for the comments in general. Our responses refer to line numbers in the revised manuscript with tracked changes and “all markup”.

We have tried to be more clear in stating the purpose of the study in the last section of the introduction. L103-111.

We do believe that an initial “simple” screening to select acids with best performance, followed by a more complex measurement to increase understanding, accuracy, and comparability to real pig house conditions is a logic and scientifically correct methodology to asses, the better acid agent for reducing greenhouse gases from pig slurry. 

2. In my opinion, the title is not appropriate because it mentions organic acids. However, most of the study deals with the inorganic acids H2SO4 and HNO3. Even in the abstract, acetic acid is only briefly mentioned once. Additionally, the authors should point out to the reader which of the acids mentioned are inorganic or organic acids.

We do believe that the part of the manuscript where the organic acids were tested are an essential part of the work, since we here rule out the possibility that combining organic acids with e.g. sulfuric acid would add significant value. Acetate was specifically mentioned in the abstract as this was most comprehensively tested in the batch screenings and is also the organic acid being most frequently mentioned in the literature. In our opinion it is redundant to mention all organic acids in the abstract. We have instead specified in materials and methods which acids are organic acids and inorganic acids. See L160-162

“Organic acids included acetic acid, lactic acid, propanoic acid, citric acid, and formic acid whereas inorganic acids used were hydrochloric acid, sulfuric acid, nitric acid, and phosphoric acid.” 

For the reasons above we do think the current title is a good description of the work presented and have therefore kept it in the revised version.

3. Why was the pH value of 5.5 in BS chosen?

We have added a phrase about this in L113-114

“In the batch screening experiments the target pH was 5.5, since this is the target pH of commercial in-house acidification systems [5].”

Other pH values could have been chosen and has also been tested in the literature. However, in commercial acidification systems in Denmark the target pH is normally 5.5 (NH4+ infarm technology, and JH acidification technology). For consistency and comparison purposes we chose this as the pH target also. 

4. There are buffer systems in the slurry which lead to an increase in the pH value after acidification. This was not taken into account or mentioned. How practically relevant are the studies from BS in which acidification was only carried out once?

We agree with the reviewer that this is important and we have added a few lines in the results sections (L310-L312) where we comment on pH increase during the batches. In addition, we included Appendix A, where the pH by the end of the experiments is plotted versus the methane production. We included some discussion about the effects of pH and how batch assays can be used and compared to commercial continuous acidification systems (L412-417)

“This discrepancy could be related to differences in acidification target pH (pH 3.8-4.8 vs 5.5) and slurry type (cattle VS pig), which would affect buffer capacity and microbial community. Importantly buffer systems in slurry can result in pH increases subsequent to acidification. Hence, the CH4 reductions reported from the batch screening experiments are not comparable to commercial-in house acidification systems with continuous pH adjustment, but instead hints to the relative efficacy of the different acids in such a system.”

5. The figures look blurred. Please improve the resolution.

The PDF builder automatically reduces resolution of figures. Therefore to see the figures clearly the reviewer should download the “high resolution” version of the figures, by clicking the download tab in the PDF. 

6. The actual data from which the mean values were formed were not provided.

We have included S4 Appendix with all the data used for making the figures. 

7. It is recommended that the manuscript is read by a native speaker before submission, as there are some linguistic and grammatical errors in the manuscript.

For example

Page 21, l. 407, have instead of has

Page 23, l. 455, was instead of were

Page 23, l. 464 ...and I would like...

We have corrected these mistakes and other grammatical errors throughout the manuscript. 

DETAILED COMMENTS

1. Page 1, l. 3, Why does it say "Introduction" after the title?

Corrected

2. Page 3, ll.47-56, This information does not add much to the understanding of the study.

We have reconsidered this part and agree that it is redundant. Therefore we have removed this paragraph and replaced it in L62-64

“Large CH4 reductions were previously observed from HNO3 acidified cattle slurry, but simultaneous increases in N2O emission attributed to denitrification activity was also detected [15,16]. “ 

In addition we shortened the paragraph about denitrification in L52-53

3. Page 4, l. 78, There is a missing space before the source [20].

Corrected

4. Page 5, l. 95, With state-of-the-art, one space is too many.

Corrected

5. Page 5, ll.97-98, It is unusual for results to already be presented in the aim in the introduction.

We included this to comply with the PLOS author guidelines for the introduction: 

“Conclude with a brief statement of the overall aim of the work and a comment about whether that aim was achieved”

6. Page 6, ll. 120-124, Why were different types of slurry used (especially for CHS experiment 1 and 2)?

The reviewer is correct in that CHS1 (CHS A) could have been conducted with the same slurry as in CHS2 (CHS B). 

However, as the slurry by the end of the experiment showed reduced methanogenic activity we chose a more potent “residual slurry” for the next CHS experiments. More activity in the slurry is necessary to see clear differences between treatments. We don’t believe this affects our conclusions since comparisons are only made within each experiment

We have added and explanation of the different in slurries in L137-139

“Due to decreasing methanogenic activity in the vacuum flushed slurry a residual slurry with higher methanogenic activity was used in CHS experiment B and C to ensure that potential differences in treatments were still detectable.” 

7. Page. 7 l. 130, Table 1 is not mentioned in the text.

This is a mistake. It has been added in L135

8. Page 7, l. 130, Why are the abbreviations for the slurry relevant? Are they even mentioned again in the manuscript?

Thanks for catching this redundancy. Abbreviations has now been removed from the table. 

9. Page 7, l. 140, Table 2 instead of Table 1, please check all cross-references again!

Corrected and checked

10. Page 8, l. 147, BS experiment 1: length 6 or 18 days; write out abbreviations such as H2SO4 or HCl in the text; explain mM in the table heading; BS experiment 3: H2SO4 better mentioned in 2nd place

Corrected: mM explained in table subheading. “6 or 18 days” added

We understand this as the reviewer prefers to replace chemical formulas with names (e.g. H2SO4 with sulfuric acid) only in the Table (not in the main text). This has been done in Table 2.

We have moved sulfuric acid to 2nd place in the table. 

We have still kept the chemical formulas in the main text of the manuscript. We were unsure if the reviewer also thought these should be changed. We could not find any requirement about this in the guidelines.

11. Page 9; ll. 161-163, Show H2SO4 scrubber and Nafion tube in Figure 1 for better understanding.

This has now been included in Fig 1.

12. Page 10, l. 165, Table 3 instead of Table 2

Corrected

13. Page 10, l. 167, The correct spelling is 20 g HNO3 (69% purity) (kg slurry)-1 (please check in the whole manuscript).

Corrected

14. Page 10, l. 168, Methane is also a greenhouse gas.

Rephrased to avoid confusion about this. See L195-199. 

15. Page 10, l. 185-187, How was the pH measurement carried out? Was the vessel opened so that air could enter? Was the sample stirred, which also led to oxygen entering and thus influenced the pH development (stronger pH increase as mentioned in other studies)?

We have added details about acidification procedure in L193-195 and general pH measurement in L216-218. 

“Acidification in all CHS experiments were carried out by alternately pipetting acid into the slurry under agitation with a magnet stirrer followed by a pH measurement. This procedure was repeated until the pH reached and remained stable at the target pH over 5 minutes.”

“For all CHS experiments the pH was measured every 3–4th day by opening the incubation flasks completely and carefully submerging the pH sensor to 1 cm above the bottom of the incubation flask without stirring of the slurry.”

When only measuring the pH (no pH adjustment) the flasks were opened completely by taking of the lid and the bulk pH was measured by submerging the pH sensor 1 cm above the bottom. The slurry was not stirred during this process. The continuous headspace gas was air, so the slurry surface was exposed to oxygen at all times during the experiment, independently of the pH measurement (described in 2.4). Therefore pH measurements was not considered to influence much on the biochemical processes in the slurry, except for breaking the pH gradient at the place of submerging. This, however, is no different from barn conditions where fresh manure is excreted to the pit, disturbing the bulk slurry. 

In the case of acidification and residual slurry acidification it was necessary to stir the manure intensely to ensure the target pH was correctly reached. 

16. Page 11, l. 188, the letters "L", "H" and "B" were not explained

This information has been added in a Table footnote. B was referring to the manure type, earlier described in Table 1. But we have now removed “B” (and other letters referring to slurry types) to avoid redundancy. 

17. Page 11, l. 188, Why were different amounts of slurry used in experiment 1 and 2?

The lower volume used was for practical reasons when acidifying the slurry, which created a lot of foam. Since the pH was decrease to 5.5 for all treatments in CHS experiment 2, more foam was expected.

We added this in L201-202

“Compared to CHS experiment A, the slurry mass was reduced (from 800 to 500 g) to avoid excessive foam formation during acidification.”

18. Page 11, Chapter 2.4 it is better to mention this chapter earlier in the material and methods section, as results have already been presented in Table 1. Alternatively, the results of the slurry characterisation could also be presented in the results chapter.

We moved up “Slurry characterization” as the first section in materials and methods. 

19. Page 12, l. 207, The abbreviation ATM was not explained

Has been written out as “atmosphere” 

20. Page 14. l. 251, Why not experiment A, B and C instead of 1, 2 and 3. A better naming of the individual sub-experiments would help the reader to understand the manuscript.

We have changed 1,2,3 to A,B, and C for batch experiments and CHS experiments throughout the manuscript

21. Page 15, ll. 259-260, It is better to mention this in the material and methods section

We removed this and included it in the M&M

22. Page 15, l. 261-262, It is better to mention this in the material and methods section

We removed this and included it in the M&M

23. Page 15, l. 265, Why is the pH value of the variant acidified with HNO3 higher than that of the control?

It could be due to denitrification, but the many processes that could occur (e.g. CO2 loss in the beginning) are blurring the picture of what caused this increase. We have written this into the manuscript L 310-312.

24. Page, 16, l. 293, one point too many at the end of the sentence

Corrected

25. Page 23, ll. 458-461, Although the authors correctly point out that a daily addition of fresh slurry makes acidification less efficient, this cannot be avoided in practice in the barn. Or do the authors have any ideas?

We are aware this is unavoidable. In the authors opinion it could mean that residual slurry acidification should have a lower target pH – perhaps even 4.5. 

We added a phrase in L489-492

“The reduced effect of residual slurry acidification compared to bulk acidification suggest that residual slurry acidification should be carried out with a lower target pH than 5.5 to maintain the inhibiting effect once new slurry is added.”

26. Fig. 2C, Are there any statistics for this illustration?

No statistics were done here since all treatments were different, even the ones with the same organic acid used different concentrations of it. Therefore we chose to show all variants in Fig 2C.

27. Fig. 3, The pH value normally increases more quickly after acidification of slurry. Presumably, the slower increase in experiment 1 can be explained by the old slurry > 2 months. This slurry is therefore not representative, especially if you think of an in-house slurry acidification.

It is difficult to say how the pH developed in the start of CHS experiment 1 (now called CHS A), since it was measured for the first time after 1 week (the line is an interpolation). We are not convinced that general statements of how fast pH increases can be made in this case where 1) HNO3 was used and at different pH values. However, the reviewer might have a point that the slurry is not completely representative of barn conditions, but within CHS A, the different treatments are still comparable. 

28. Fig. 4A, Why do the CH4 emissions decrease on day 7/8 and increase again on day 12/13 after the interruption of the measurement period? The CO2 concentration is also lower between days 6-8? Could the H2SO4 scrubber have had an influence?

This is a mistake in the data treatment and we are glad the reviewer caught this mistake. 

The CH4 increase and decrease is due to the scrubber being applied and removed. Since values are averaged over 12 h intervals to make plots more easily interpretable (now specified in the statistics section). The concentration on Fig 4A was influenced also outside the intervals of the scrubber. We have corrected this in the figure now to avoid confusion of whether the CH4 is increasing/decreasing or if it is due to the scrubber being used or not. The decrease in CH4 emission by the end of the experiment (in the untreated sample) is probably due to substrate depletion or depletion of easily degradable substrate. 

For CO2, we have since found that some VOCs may have interfered with CO2 concentrations and therefore only periods with the scrubber + nafion tube applied is now shown. Our background measurements of CO2 and air shows expected CO2 values. We have cleared this out in the new Fig 4 and explained the interference in L196-199

“In CHS experiment A Nitrous oxide was measured continuously over two weeks, but CH4 and CO2 were only measured periodically when a Nafion filter was applied to remove NH3 and possibly other volatile organic compounds, that caused spectral interference with with CH4 and CO2.”

Reviewer #2: This study studied GHG emissions from acidified pig slurry with different organic and inorganic acids. Results showed HNO3 could result in N2O emissions and therefore not suitable. This study design followed a straightforward while rigorous manner by applying firstly a large amount of screening tests, followed by continuous headspace experiments. I find the results interesting. I have only some minor comments.

Thanks for the comments in general. Our responses refer to line numbers in the revised manuscript with tracked changes and “all markup”.

1. It was believed the N2O was due to denitrification of NO3-, was this N2O production/accumulation due to acidic condition or inherent partial denitrification?

This is a good question. Since low pH inhibit N2O-reductase (N2O to N2) we would expect low N2O production with higher dose of HNO3 (lower pH). However, as seen from Fig 4 the lower pH also reduced CO2 production, which indicates that microbial activity was inhibited as well. We believe therefore that the inhibition of microbes able to carry out partial denitrification was inhibited at acidic conditions. 

In L442-446 we added the following

“It has been shown that enzymatic N2O reductase activity is significantly reduced at low pH [29], hereby decreasing the enzymatic reduction of N2O to N2. However, that fact that CO2 production was reduced below pH 6, suggests that general inhibition of heterotrophic activity (including heterotrophic denitrifiers) is a more plausible explanation for the reduced N2O emission below pH 6.”

2. There is no synergy between inorganic and organic acid. Is there synergy between H+ and NO3-/SO4- (as electron acceptors)?

This is a good question and we don’t believe that our data can support any conclusion about this. It has previously been shown that sulfate inhibits methanogenesis it self – possibly due to SO4 being preferable electron acceptor compared to CO2 for methanogenesis. 

In order to answer the question we should have conducted experiments with pure SO4 and NO3 addition and compared to that of H2SO4 and HNO3. However, this was out of the scope, but certainly an interesting point. 

3. You can also discuss the economic implications of these investigated acids.

Good point. We have, however, chosen not to go into the economics of acidification as a good cost benefit analysis is beyond the scope.

Reviewer #3: This study investigated the effect of pig slurry acidification with a range of organic and inorganic acids with respect to their CH4 inhibitor potential in a number of batch experiments. There are some issues with this manuscript which need to be addressed before it can be accepted to publish, and they are listed below.

Thanks for the comments in general. Our responses refer to line numbers in the revised manuscript with tracked changes and “all markup”.

1. Line 97, the subscript should be corrected. Please check similar problems in the text.

Corrected

2. In Table 1, could the difference in original TAN between the slurries affect the batch experiments?

For the batch experiments the same slurry was used, so there is not any difference in the TAN concentration here (Table 1 in first data row). If the reviewer refers to the continuous headspace experiments, there was a difference between CHS1 vs CHS2 + CHS3. However, in these experiments the methanogenic potential as well as dry matter and pH was also different. We believe it is more likely that other factors than TAN affected the experiments.

In “Astals et al. 2018. Characterising and modelling free ammonia and ammonium inhibition in anaerobic systems” TAN inhibition was studied in pig slurry and using their equations it is unlikely that TAN was inhibiting much on methanogenesis in our study. 

In any case CHS experiments were compared only within and not among experiments, and therefore any possible TAN inhibition would be equally effecting the experiments. 

3. Line 293, a redundant full stop.

We combined the sentences into one

4. Line 339, 344 and 348. I don’t know if the references format meet the requirements of PLOS ONE?

This has been corrected. 

5. Line 386-387, have the denitrification activity been measured? If not, it is not appropriate to state like that.

We measured it indirectly by measuring N2O production. Previous studies with HNO3 addition to slurry found that denitrification was the main source of N2O. See Oenema, O. and Velthof, G.L. 1993 (cited in manuscript). 

However we softened the sentence by replacing “accounted for“ to “corresponded to” in L437 

6. Line 446-447, “It is likely that high concentrations of VFAs present prior to acidification reduce the efficacy of organic acid treatment.” The evidence to support the conclusion should be clarified in the text.

This conclusion, which is still soft (using the word “likely”), is based on L389-401 under the discussion of the batch experiments.

---

## [Editor Report · Decision Letter 1]

14 Apr 2022

Reducing greenhouse gas emissions from pig slurry by acidification with organic and inorganic acids

PONE-D-22-02024R1

Dear Dr. Dalby,

We’re pleased to inform you that your manuscript has been judged scientifically suitable for publication and will be formally accepted for publication once it meets all outstanding technical requirements.

Kind regards,

Xueming Chen

Academic Editor

PLOS ONE
---

## [Editor Report · Acceptance letter]

26 Apr 2022

PONE-D-22-02024R1 

Reducing greenhouse gas emissions from pig slurry by acidification with organic and inorganic acids 

Dear Dr. Dalby:

I'm pleased to inform you that your manuscript has been deemed suitable for publication in PLOS ONE. Congratulations! Your manuscript is now with our production department. 

Kind regards, 

on behalf of

Dr. Xueming Chen 

Academic Editor

PLOS ONE